# Prevalence of prolonged transitional neonatal hypoglycemia and associated factors in Ethiopia: A systematic review and meta-analysis

Solomon Demis Kebede[ID][1]*, Amare Kassaw[1], Tigabu Munye Aytenew[2], Kindu Agmas[3], Demewoz Kefale[1]

1 Department of Pediatrics and Child Health Nursing, College of Health Sciences, Debre Tabor University, Debre Tabor, Ethiopia, 2 Department of Nursing, College of Health Sciences, Debre Tabor University, Debre Tabor, Ethiopia, 3 Departments of Pediatrics and Child Health, Debre Tabor Comprehensive Specialized Hospital, Debre Tabor, Ethiopia

* solomondemis@gmail.com

**Data Availability Statement:** All relevant data are within the paper and its Supporting information files.

## Abstract

### Introduction

Most neonates experience transient hypoglycemia, which typically responds well to treatment and is associated with a favorable prognosis. However, hypoglycemia persisting beyond 48 hours, termed prolonged transitional Neonatal hypoglycemia (PTNHG), can result in abrupt neuronal injury and long-term neurodevelopmental impairments. Identifying its prevalence and associated risk factors is critical to inform clinical practices and improve neonatal outcomes.

### Methods

A weighted inverse-variance random-effects model was employed for the analysis. Heterogeneity among the studies was assessed using a forest plot, $I^2$ statistics, and Egger's test. Data extraction was conducted from May 20 to May 27, 2023, for studies published since 2020. A random blood sugar (RBS) concentration of <47 mg/dL measured 48–72 hours after birth was used to define PTNHG. Eight studies comprising a total of 3686 neonates were included in the analysis.

### Results

The pooled prevalence of PTNHG was 19.71% (95% CI: 16.85–22.56) with substantial heterogeneity ($I^2$ = 79.20%, P < 0.001). Subgroup analysis revealed that PTNHG prevalence was similar for studies with sample sizes >400 and ≤400, at 18% (95% CI: 15–22) and 21% (95% CI: 17–26), respectively. Similarly, prevalence estimates were comparable when using RBS thresholds of <47 mg/dL (21%; 95% CI: 16–27) and <40 mg/dL (18%; 95% CI: 15–22). Significant factors associated with PTNHG included preterm birth (AOR = 3.31; 95% CI: 2.57–4.04), hypothermia (AOR = 3.41; 95% CI: 2.19–4.62), being an infant of a

**Funding:** The authors received no specific funding for this work.

**Competing interests:** The authors have declared that no competing interests exist.

**Abbreviations: CS**, Cesarean Section; **GDM**, Gestational Debates Mellitus; **HIE**, Hypoxic Ischemic Encephalopathy; **IDM**, Infant of Diabetics' Mother; **PTNHG**, Prolonged Transitional Neonatal Hypoglycemia; **SGA**, Small for Gestational Age; **SVD**, Spontaneous Vaginal Delivery; **VLBW**, Very Low Birth Weight.

diabetic mother (IDM) (AOR = 4.71; 95% CI: 2.15–7.26), delayed breastfeeding initiation beyond one hour (AOR = 3.26; 95% CI: 2.03–4.49), and pathological jaundice (AOR = 2.37; 95% CI: 1.91–2.84).

## Conclusions

Nearly one-fifth of hospitalized neonates experienced PTNHG. Fortunately, most of the associated risk factors were modifiable. Prioritizing early breastfeeding initiation, particularly in cesarean section deliveries and IDM cases, and integrating PTNHG management into national NICU guidelines could significantly reduce the burden of neonatal hypoglycemia.

## Trial registration

**Prospero ID**: CRD42023424953. https://www.crd.york.ac.uk/prospero/display_record. php?ID=CRD42023424953.

## Introduction

Neonatal hypoglycemia is defined as low blood glucose concentration below 40–50 mg/dL as per World Health Organization (WHO) and American Academy of Pediatrics (AAP)[1,2]. There are two types of neonatal hypoglycemia: transient and prolonged. Most neonates have transient hypoglycemia, which responds to treatment and is associated with a good prognosis compared with persistent hypoglycemia. Hypoglycemia that occurs after 48 hours of age is termed prolonged transitional hypoglycemia, the clinical manifestations of which are generated in the short term, with abrupt injury and neurodevelopmental sequelae occurring over the long-term [3,4].

Observations have shown that healthy infants experience transient hypoglycemia as a part of the normal adaption to extrauterine life, with a decline in blood glucose concentrations to values as low as 20 to 25 mg/dL in the first two hours of life[3]. Neonatal hypoglycemia persisting beyond 48 hours after birth might be prolonged transitional hypoglycemia[4].

Deviations from the normal transition can occur as a result of a wide range of factors predisposing individuals to neonatal hypoglycemia. The brain uses glucose primarily to meet its metabolic requirements. Therefore, neonatal hypoglycemia is a significant medical emergency in neonatal units. The signs and symptoms are often nonspecific. Some neonates are completely asymptomatic. Many cases, therefore, remain undiagnosed. The diagnosis of neonatal hypoglycemia requires a high index of suspicion since its presentation is different. Early diagnosis and treatment could prevent abrupt signs and symptoms, long-term complications, and death [2,5,6].

The neonatal hypoglycemia is still a severe concern because it causes neurological defect, an increased risk of mortality, and other complications in neonates and families. A study conducted by Shah R et al., (2019) revealed that the risk of a specific cognitive defect increased by 2-3-fold in early childhood. Neonatal hypoglycemia had odds of 3.2 for neurodevelopmental impairment and 2.04 for low literacy at school in late childhood [7]. And, neonates with hypoglycemia would developed cerebral palsy at 8% in their childhood[8].

Moreover, a study in Tanzania showed that hypoglycemia was a major cause of neonatal mortality, contributing to 20% of all causes. Other family-related burdens secondary to neonatal hypoglycemia admission include increased costs to the family and separation from the

mother[9]. Thus, detection and treatment of these groups of neonates are targeted to reduce neonatal death, mainly in low-resource countries [10–13]. According to some studies, neonatal hypoglycemia affects up to approximately 4–5 per 1000 term infants [5,6]. A blood glucose level (RBS) below 47 mg/dL (2.6 mmol/L), if present during the 5 days during the first two months of life, was significantly correlated with acute seizure activity, abnormal neuromotor manifestations, and impaired intellectual performance at 18 months of age[14].

Despite progress over the past two decades, the age distribution of the mortality of children and young adolescents shows that the highest risk of death occurs during the neonatal period (the first 28 days of life), during which preventable causes accounted for 2.5 million out of 6.2 million deaths of children and young adolescents in 2018 alone[3,15]. The neonatal mortality rate was estimated to be 18 deaths/1,000 live births globally[3,15]. The probability of dying after the first month and before reaching age 1 was 11 per 1,000 live births [3,15].

According to the Ethiopian Demographic Health Survey (EDHS), Ethiopia has one of the highest neonatal mortality rates in the world (29 per 1000 live births), which represents poor progress in reducing the rate of neonatal mortality from 2000 to 2016[16]. The report of previous studies on the prevalence of neonatal hypoglycemia and its risk factors were inconsistent in findings, and it is the first by its nature on prevalence of prolonged transitional hypoglycemia and associated factors in Ethiopia. The increased prevalence of prematurity and low birth weight make such a review vital for the formulation of prevention and management to neonates in specific and under-five children in general.

## Methods

The results of this review were reported based on the Preferred Reporting Items for Systematic Review and Meta-analysis (PRISMA) statement guidelines[17](S1 File). Data was extracted from 20–27 May 2023 for studies since 2020. The protocol is registered in the PROSPERO database (PROSPERO ID: CRD42023424953; https://www.crd.york.ac.uk/prospero/myprospero.

### Search strategy and information source

The adapted PICO format was used to explicitly review the literature and clarify the specifications of the inclusion and exclusion criteria. The adapted PICO comprises population (P), exposure (E), outcome (O), and context (setting as described below).

1. **Population**: All neonates irrespective of gestational age and birth weight

2. **Exposure**: Associated factors, predictors, risk factors

3. **Outcome**: prolonged transitional neonatal hypoglycemia

4. **Context (Setting):** Ethiopia.

Hence, by using this adapted PICO format, the following review questions were developed for the search for data.

1. What is the incidence of prolonged transitional neonatal hypoglycemia in Ethiopia?

2. What are the factors associated with prolonged transitional neonatal hypoglycemia in Ethiopia?

Based on the aforementioned review PICO format and questions, primary studies were identified from MEDLINE, Scopus, Embase, Science Direct, Google Scholar, the African Journal Online, the Addis Ababa University repository, and the Haramaya University repository.

The core search terms and phrases used were "prevalence", "incidence", "epidemiology", "proportion", "magnitude", "burden", "persistent", "prolonged", "late newborn", "recurrent", "low blood glucose", "predictors", "risk factors", "associated factors" and "Ethiopia". The search strategies were developed using different Boolean operators. Notably, to fit the advanced MEDLINE database, the following search strategy was applied on April 8, 2023: [(prevalence[MeSH Terms]) OR incidence[MeSH Terms]) OR proportion [MeSH Terms]) OR epidemiology[MeSH Terms]) OR magnitude[MeSH Terms]) OR burden[MeSH Terms]) AND predictors [MeSH Terms]) OR risk factors [MeSH Terms]) OR associated factors [MeSH Terms]) OR neonatal hypoglycemia [MeSH Terms]) OR newborn hypoglycemia [MeSH Terms]) OR low blood glucose in newborns [MeSH Terms]) OR low random blood glucose level in neonates [MeSH Terms]) AND (Ethiopia)].

## Study selection

Primary studies downloaded and retrieved were exported to Mendeley Desktop 1.19.8 reference manager software to remove duplicate studies. The study selection process was held in two stages. First, the title and abstract were screened, and second, a full-text review was performed. Two independent reviewers (DK and AK) screened the title and abstract. Disagreements were resolved based on established article selection criteria, and the third author (SDK) resolved disagreements to decide whether the study was eligible for inclusion if the quality appraisal score was 5 and more out of 9. Two independent authors (TMA and KA) subsequently reviewed the abstracts and full texts of the eligible articles.

## Eligibility criteria

**The inclusion criteria for patients were as follows**: articles that reported the prevalence of prolonged/persistent transitional neonatal hypoglycemia in general and/or at least one associated factor and published in English and studies conducted only in Ethiopia.

**The exclusion criteria for patients were as follows:** forms of hyperinsulinism or other diseases causing hypoglycemia such as inborn errors of metabolism were excluded. Moreover, articles without full-text information and qualitative studies were not reviewed.

## Critical appraisal and quality assessment

We used the quality appraisal criteria of the Joanna Briggs Institute (JBI)[18]. The tool consists of nine major items. The first item is appropriate for the sample frame. The second is the appropriate sampling technique. The third is the adequacy of the sample size. The fourth is a description of the study subjects and settings. The fifth is enough coverage of the data analysis. The sixth is the validity of the method for identifying the condition. The seventh item is a standard and reliable measurement for all participants. The eighth is the appropriateness of the statistical analysis. The last item is adequacy and management of the response rate. Studies were considered low risk when they fit 5 or more quality assessment checklists. Two independent authors (TMA and KA) appraised the quality of the studies. Disagreements were resolved by the interference of a third reviewer (SDK). The quality scores for the primary studies were reported (S2 File).

## Data extraction

The data were determined to be extracted based on two criteria: 1. Clear and consistent operational definitions for the dependent variable (persistent neonatal hypoglycemia) and 2. These variables are statistically associated with the outcome variable reported by the AOR. Two

authors extracted the data using the standardized format of the MS Excel spreadsheet (S3 File). The name of the first author and year, study region, study area, study design, random blood glucose (RBS) cutoff point, sample size, prevalence, and odds of associated factors were extracted from 20–27 May 2023. Whenever variations were observed, the phase was repeated. If discrepancies between the data were resolved, a third reviewer was consulted. After reporting the inconsistency of the data in the primary study, data transformation was conducted.

### Outcome measures

**Prolonged (persistent) transitional neonatal hypoglycemia (PTNHG):** refers to neonates with a random blood sugar (RBS) concentration of <47 mg/dL measured 48–72 hours after birth using a bedside random blood glucose analysis performed on a capillary sample [3,4].

### Statistical analysis

The required data were collected using the Microsoft Excel 2013 workbook. Then, STATA version 17 statistical software was used for the meta-analysis. Publication bias was objectively checked using Egger's regression test analysis[19]. The heterogeneity of the studies was quantified using the I-squared statistic [20]. Both pooled prevalence analysis and pooled effect of associated factors were conducted using a weighted inverse variance random-effects model [21]. Subgroup analysis was performed by diagnostic method and sample size to overcome the inflation of the pooled effect from the inclusion of studies.

## Results

### Literature search

The search strategy retrieved 64 articles electronically and manually, of which 18 duplicate articles were identified and removed. Among the 46 screened articles, 46 were from PubMed (n = 19), Google Scholar (n = 23), and other databases of the Addis Ababa University repository (n = 2), the Haramaya University database (n = 1), and the manuscript (n = 1). The full texts of nineteen articles were reviewed, and eight studies were included in the review and meta-analysis (Fig 1).

### Characteristics of the included studies

Eight cross-sectional studies with a pooled sample size of 3,686 neonates were included in this meta-analysis. Two studies were found in each part of Ethiopia, namely, at Central, Eastern, Northwest and Northern [5,22–28]. Four studies used an RBS cutoff of <40 mg/dl for the diagnosis of prolonged transitional neonatal hypoglycemia [22,23,27,28]. The remaining four studies used an RBS cutoff point of less than 47 mg/dl for the diagnosis of PTNHGs [5,25,26,29]. The maximum sample size was 769, and the minimum sample size was 196 among the reviewed studies [27,29]. The studies included in this systematic review and meta-analysis had no considerable risk. Therefore, all the studies were considered. We assessed the studies with a JBI quality appraisal checklist for systematic review [30]. According to this appraisal checklist, none of the included studies were of poor quality and had a quality score ≥6 (Table 1).

### Meta-analysis

The pooled prevalence of prolonged transitional neonatal hypoglycemia was reviewed, and the absence of publication bias was assessed with Egger's regression test analysis (p = 0.068), which showed no publication bias (Table 2).

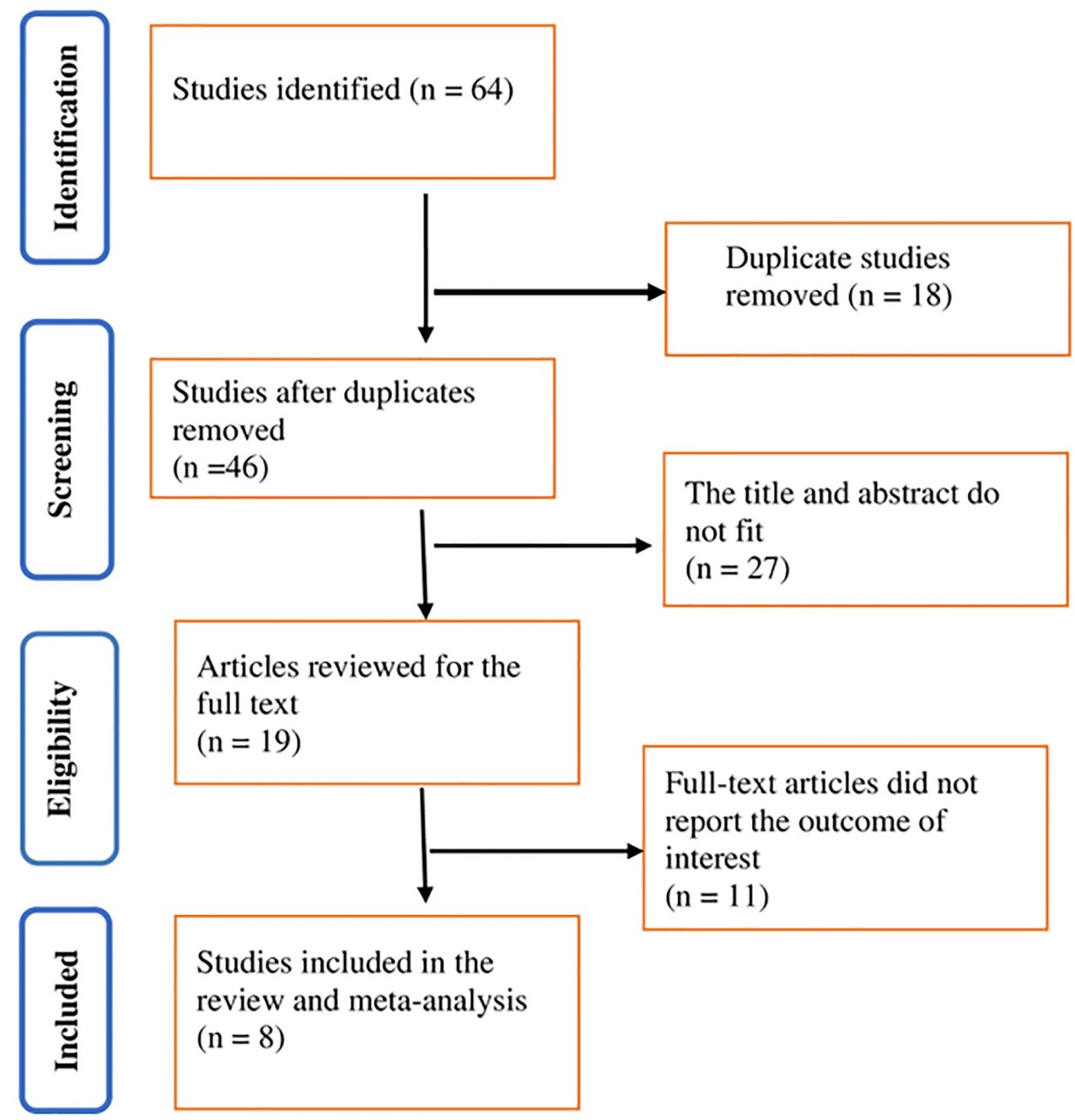

**Fig 1. An adapted PRISMA 2020 flow diagram for prevalence of prolonged transitional neonatal hypoglycemia and associated factors in Ethiopia, 2023.**

The pooled prevalence of PTNHG from eight studies [22–29] was 19.71 (95% CI: 16.85–22.56; $I^2$ = 79.20, P<0.001) (Fig 2).

## Subgroup analysis

Subgroup analysis was performed by using the sample size and the RBS cutoff point to diagnose prolonged transitional neonatal hypoglycemia (PTNHG). However, there was no effect

**Table 1. Characteristics and quality status of the studies included, 2023.**

| Author and year | Parts of Ethiopia | Capillary blood RBS cut point | Sample size | Incidence (%) of PNHG | JBI quality status |
|---|---|---|---|---|---|
| Bogale et al., 2021 [28] | Northern (UOG) | <40 mg/dl | 399 | 13.53 | Low risk |
| Demis et al., 2022 [26] | North-central (DTCSH) | <47 mg/dl | 400 | 23.50 | Low risk |
| Fantahun & Nurussen, 2021 [29] | Central St. (PHMMC) | <47 mg/dl | 196 | 25.00 | Low risk |
| Kasaye et al., 2021 [22] | Central (Addis Ababa) | <40 mg/dl | 317 | 20.82 | Low risk |
| Sertsu et al., 2022 [23] | Eastern (HFSTH) | <40 mg/dl | 698 | 21.20 | Low risk |
| Yohannes et al., 2021 [24] | Eastern (Harar and Dire Dawa) | <47 mg/dl | 640 | 14.68 | Low risk |
| Chanie et al., 2023 [25] | North-central (DTCSH) | <47 mg/dl | 267 | 23.60 | Low risk |
| Demisse et al., 2017 [27] | Northern (UOG) | <40 mg/dl | 769 | 18.46 | Low risk |

UOG = University of Gondar, St. PHMMC = Saint Paul's Millennium Medical College, DTCSH = Debre Tabor Comprehensive Specialized Hospital, Data extracted date was on 20-27/05/2023, Data extractors were SDK, TMA, DK and KA.

**Table 2. Egger's test indicating no publication bias.**

| Std.-Eff. | Coef. | Std. Err | t | p>t | 95% confidence interval | |
|---|---|---|---|---|---|---|
| **Slope** | 5.21962 | 2.177999 | 2.40 | 0.054 | -.1097508 | 10.54899 |
| **Bias** | 9.567008 | 4.312786 | 2.22 | **0.068** | -.9859986 | 20.12001 |

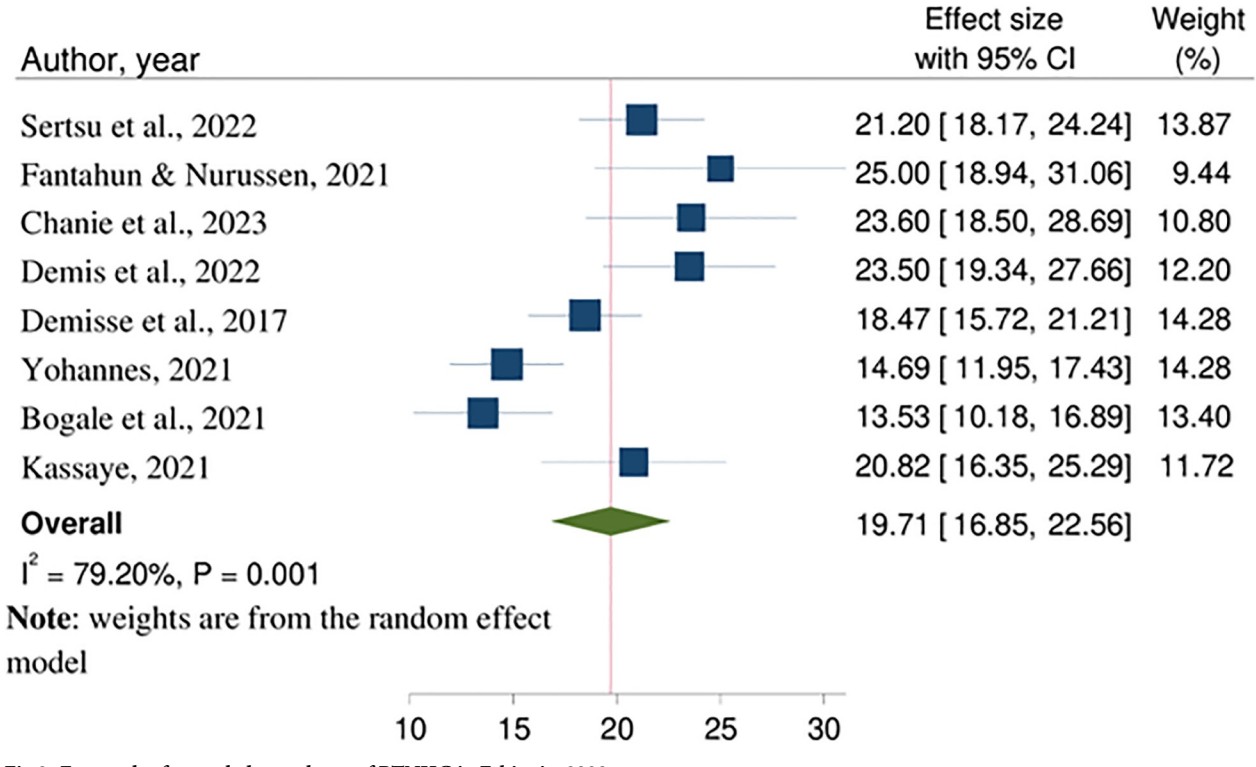

**Fig 2. Forest plot for pooled prevalence of PTNHG in Ethiopia, 2023.**

**Table 3. Subgroup analysis by sample size category and random blood sugar cutoff point for PTNHG and associated factors in Ethiopia, 2023.**

| Subgrouping by variables | Authors | Pooled prevalence at (95% CI) | I$^2$(P value) |
| --- | --- | --- | --- |
| Sample size >400 | [23,24,27] | 18(15–22) | 78%(0.01) |
| Sample size ≤400 | [22,25,26,28,29] | 21(17–26) | 79.67%(0.00) |
| RBS<40 mg/dL | [22,23,27,28] | 18(15–22) | 73.83%(0.01) |
| RBS<47 mg/dL | [24–26,29] | 21(16–27) | 85.34%(0.00) |

Data extracted date was on 20-27/05/2023, Data extractors were SDK, TMA, DK and KA.

on reducing the heterogeneity of the heterogeneity (I$^2$). According to the three studies of pooled effects, the prevalence with sample sizes >400 was 18% (15–22), and the prevalence with sample sizes ≤ 400 was 21% (17–26). The pooled effect of the RBS cutoff point for the diagnosis of PTNHG by (<40 mg/dl) were 18% (15–22), and by (<47 mg/dl) was 21% (16–27) (Table 3).

## Factors associated with PTNHG

Based on this review, the factors associated with persistent neonatal hypoglycemia identified by pooling two or more studies were preterm birth, neonatal hypothermia, neonatal birth from a chronic diabetic mellitus mother, breastfeeding initiation after an hour, and pathological jaundice.

According to previous studies [23,26], neonates who were delivered prematurely were three more likely to develop prolonged transitional neonatal hypoglycemia than neonates who were born at term in Ethiopia (AOR = 3.31; 95% CI = 2.57–4.04). A pooled analysis of four studies [23,25,26,29] revealed that neonates with hypothermia were 3.41 times more likely (AOR = 3.41; 95% CI = 2.19–4.62) to develop PTNHG than their normotensive counterparts were. There were nearly five times more neonates whose mothers had known chronic DM (AOR = 4.71; 95% CI = 2.15–7.26) than did those whose mothers did not have known chronic DM [22,23,25].

According to previous studies [22,23], neonates who had been breastfeeding after an hour were three times more likely to be hypoglycemic than neonates who had been breastfeeding within an hour (AOR = 3.26; 95% CI = 2.03–4.49). By pooling of two studies[24,26] revealed that the incidence of pathological jaundice in neonates was almost twofold-fold greater (AOR = 1.70; 95% CI = 1.21–2.99) than that in their counterparts without pathological jaundice. One study [23] revealed that the odds of neonatal sepsis were nearly three times greater than that of neonates with no sepsis, and the odds of neonates with very low birth weight (VLBW) were four times greater than those of neonates with normal birth weight(NBW) (AOR = 2.61; 95% CI = 1.68–3.54) and (AOR = 4.01; 95% CI = 1.85–9.87), respectively (Table 4).

According to a previous study [25], for neonates with one or more episodes of seizure and asphyxia stage II or III HIE, PTNHG had odds of nearly five times and five times greater than their counterparts (AOR = 4.70; 95% CI = 3.78–5.62) and (AOR = 5.10; 95% CI = 4.18–6.02), respectively. Other single studies (Kasaye, 2021) [22] showed that neonates whose mothers had GDM or preeclampsia during their latest pregnancy were approximately eight or two times more likely to have GDM or preeclampsia, respectively (AOR = 7.79; 95% CI = 5.87–9.71) and (AOR = 2.44; 95% CI = 1.65–3.23), than their counterparts were. In contrast, a study by [26] revealed that neonates delivered via SVD were 28% less likely to develop PTNHG than were those delivered via CS (AOR = 0.72; 0.35–1.79).

**Table 4. The pooled effect of factors associated with PNHG in Ethiopia, 2023.**

| Factors | Author and year | Pooled OR (95% CI) | I²(P value) |
|---|---|---|---|
| Preterm birth | (Sertsu et al., 2022) [23] | 3.31(2.57–4.04) | 0.01%(0.001) |
| | (Demis et al., 2022) [26] | | |
| Neonatal Hypothermia | (Sertsu et al., 2022) [23] | 3.41(2.19–4.62) | 97.96%(0.001) |
| | (Fantahun and Nurussen, 2021) [29] | | |
| | (Chanie et al., 2023) [25] | | |
| | (Demis et al., 2022) [26] | | |
| Infants of diabetic mother (IDM) | (Sertsu et al., 2022) [23] | 4.71(2.15–7.26) | 94%(0.001) |
| | (Chanie et al., 2023) [25] | | |
| | (Kasaye et al., 2021) [22] | | |
| Breastfeeding after one hour | (Sertsu et al., 2022) [23] | 3.26(2.03–4.49) | 52.80%(0.001) |
| | (Kasaye et al., 2021) [22] | | |
| Pathological jaundice | (Demis et al., 2022) [26] | 2.37(1.91–2.84) | 0.01%(0.001) |
| | (Yohannes et al., 2021) [24] | | |

Data extracted date was on 20-27/05/2023, Data extractors were SDK, TMA, DK and KA.

## Discussion

This systematic review and meta-analysis aimed to assess the pooled incidence of PTNHGs and associated factors in Ethiopia. The pooled prevalence was 19.71 (95% CI: 16.85–22.56), which was comparable to that reported in studies in Iran Tehran [31], China Shanghai [32] and the USA [33], which had a prevalence of 15.50% and 16.90%, respectively, and a range of 8–30%, respectively.

On the other hand, studies from Pakistan used RBS cutoff values <50 mg/dl [34], and on the basis of RBS cutoff values <40 mg/dl and 47 mg/dl [35], it was found that the incidence of neonatal hypoglycemia was much lower than that in this review (0.4% and 3.4–12.10%, respectively). This might be because this review utilized the operational definition of persistent neonatal hypoglycemia, and the aforementioned literature included all types of neonatal hypoglycemia. Moreover, the variation in sample size and socio-demographic variables might contribute to this difference.

According to our subgroup analysis based on the RBS cutoff for diagnosing PTNHG, the estimated pooled prevalence obtained by pooling the studies with an RBS cutoff<47 mg/dl was 21% greater (95% CI: 16–27) than that obtained by pooling the studies with an RBS cutoff <40 mg/dl 18% (95% CI: 15–22). A possible explanation might be the difference between the RBS cutoff points; therefore, <47 mg/dl is associated with a greater probability that more neonates would have RBS measurements within that range. This result is supported by studies in Israel in which the incidence of neonatal hypoglycemia substantially differed between the RBS cutoff <40 mg/dl and <47 mg/dl at 3.4% and 12.10%, respectively [35]. Similarly, there was a difference in the pooled estimate of PTNHG between the primary studies with a sample size >400 at 18% (95% CI: 15–22) and those with ≤400 at 21% (95% CI: 17–26). This difference can be explained by the fact that studies with sample size variation affected the outcome of the data analysis. Hence, a sample size >400 had more strength in determining the true value statistically in the regression analysis than a sample size ≤400.

This meta-analysis revealed that preterm infants were 3 times more likely to develop PTNHG than term neonates were. This could be because preterm neonates are uniquely predisposed to hypoglycemia of any type and its related complications due to their limited

glycogen and fat stores, through which body thermoregulation can be maintained. Premature neonates' inability to generate new glucose via gluconeogenesis pathways affects their glucose utilization in metabolic processes. This, in turn, requires greater metabolic demands due to having a relatively larger brain than term neonates because they are unable to respond to a counter regulatory mechanism to hypoglycemia compared to term neonates [23,36,37].

Similarly, another factor associated with PTNHGs was neonatal hypothermia, which increased the odds of PTNHG development by 3-fold compared with that of their counterparts with normothermic neonates. The scientific evidence was congruent with these findings in that neonatal hypothermia predisposes to neonatal hypoglycemia. Neonates exhibit a metabolic response to cooling that involves chemical but no shivering thermogenesis via the sympathetic nerve discharge of norepinephrine in brown fat. This reaction increases the metabolic rate and oxygen consumption 2-3-fold by activating glycogen stores, thereby causing transient hypoglycemia as a compensatory mechanism and persistent hypoglycemia as a disorder, as impaired body glucose hemostasis results in metabolic acidosis and increased risk of death [38–42].

Neonates who gave birth from mothers with chronic diabetes mellitus were nearly five times more likely to develop PTNHG than were those from mothers without chronic diabetes mellitus. The justification for this finding is that an infant of a diabetic mother (IDM) is more likely to have a period of low blood sugar shortly after birth and during the first few days of life.

This might be because neonates obtain more sugar than needed from the mother in the uterus before delivery. As such, neonates have higher insulin levels than needed after birth, and insulin then lowers their blood sugar level, allowing them to adjust to a normal RBS after birth for several days [43–45].

Regarding breastfeeding initiation time, the odds that neonates who initiated breastfeeding after an hour were approximately three times greater (AOR = 3.26; 95% CI = 2.03–4.49) of developing PTNHG than were those who breastfed within an hour after delivery [22,23]. This evidence, combined with the pooled ORs estimated, was strongly reflected in the results of a single study indicating that neonates who delivered via spontaneous vaginal delivery(SVD) were 72% less likely (AOR = 0.72; -0.35–1.79) to develop PTNHG than were those who delivered via cesarean section delivery (CS) [26]. This is justified by the fact that breast milk is the ideal food for infants that protects against neonatal hypoglycemia. The initiation of breastfeeding within an hour is the mainstay for preventing persistent neonatal hypoglycemia, as per WHO recommendations. However, neonates who delivered via the CS were 2.01 times more likely to develop neonatal hypoglycemia. Hence, any cause that delays breastfeeding within one hour, including CS delivery, should be considered. A delay in breastfeeding results in neonatal hypoglycemia because breast milk provides all the energy and nutrients that the infant needs for the first months of life and continues to provide up to half or more of a child's nutritional needs [1,46–48].

On the other hand, neonates with pathological jaundice had approximately 2.37 greater odds (AOR = 2.37; 95% CI = 1.91–2.84) than did those without pathological jaundice [24,26]. These findings are comparable to those of other studies in that neonates with hyper-bilirubinemia were 1.81 and 1.93 times more likely to develop neonatal hypoglycemia than were those with normal bilirubin test values. In another study, 8.80% of neonates treated with phototherapy developed neonatal hypoglycemia [48,49].

This might be because the relationship between neonatal hypoglycemia and neonatal hyper-bilirubinemia in pathological jaundice might be mutually influenced by the fact that OGTT 2-h glucose levels affect neonatal hypoglycemia. Abnormal postprandial glucose levels suggested impaired glucose tolerance and β-cell dysfunction on a physiological basis due to

jaundice. As a result, neonates with jaundice are affected by the physiological process of gluco-neogenesis to maintain a norm-glycemic state.

According to the findings from a single primary study, patients diagnosed with early neonatal sepsis had greater odds of developing neonatal hypoglycemia (AOR = 2.61; 95% CI = 1.68–3.54). These findings were supported by the findings of the literature that early onset of neonatal sepsis (EONS) with neonatal hypoglycemia was approximately 8.7% and 9.6%, respectively, among neonates admitted for sepsis.

This is because neonates with neonatal sepsis tend to refuse to breastfeed, resulting in low blood sugar. Similarly, increased metabolic demand is caused by low glucose levels in sepsis and hypothermia [50,51]. According to a previous study [25], for neonates with one or more episodes of seizure and asphyxia stage II or III HIE, PTNHG had odds of nearly five times and five times greater than their counterparts (AOR = 4.70; 95% CI = 3.78–5.62) and (AOR = 5.10; 95% CI = 4.18–6.02), respectively. This increased risk for hypoglycemia might be due to a lack of storage in combination with high insulin levels, which predisposes patients to hypoglycemia. Perinatal asphyxia (PNA) and perinatal seizures secondary to PNA increase the risk of hyper-insulinism in the neonatal period because of the use of anaerobic metabolism to maintain blood glucose concentrations. Anaerobic metabolism is used to maintain blood glucose concentrations [52]. Finally, a study [22] showed that neonates with gestational diabetes mellitus (GDM) or preeclampsia during their latest pregnancy were approximately eight times and two times more likely, respectively (AOR = 7.79; 95% CI = 5.87–9.71) and (AOR = 2.44; 95% CI = 1.65–3.23), than their counterparts were. This can be explained by the fact that the most common complication in infants born to GDM mothers is hypoglycemia. This is due to glucose freely passing through the placenta; therefore, maternal hyperglycemia associated with GDM results in elevated glucose levels in the fetus, causing excess fetal insulin production, termed hyper-insulinism[53].

## Conclusions

Prolonged (Persistent) neonatal hypoglycemia is quite common in Ethiopia. The pooled prevalence estimate of PTNHG from eight studies was nearly ten percent. Neonates who delivered prematurely, neonates with hypothermia, neonates of diabetic mothers (IDM), neonates who had not been breastfeeding after an hour and neonates with pathological jaundice had higher odds of developing PTNHG than did their counterparts. According to single primary studies, neonatal sepsis, very low birth weight(VLBW), neonates with one or more episodes of seizure, and asphyxia stage II or III HIE, neonates of GDM mothers had higher odds of developing neonatal hypoglycemia. In contrast, neonates delivered by SVD were protected against neonatal hypoglycemia. As per this review, the pooled prevalence of PTNHGs was significant. Several associated factors were pooled so that intervention strategies would be needed to reduce and prevent their occurrence while treating neonates in neonatal intensive care units in Ethiopia. Thus, the team of this review strongly advised health care professionals, including nurses, doctors, and others working in the NICU, to monitor and regulate these and other determining variables that predispose patients to neonatal hypoglycemia, particularly the persistent type.

## Supporting information

**S1 File. PRISMA checklist.**
(DOCX)

**S2 File. JBI quality appraisal criteria.**
(DOCX)

**S3 File. Data of the review and meta-analysis.**
(XLSX)

**S4 File. Studies identified in the literature search.**
(DOCX)

## Author Contributions

**Conceptualization:** Solomon Demis Kebede.

**Data curation:** Solomon Demis Kebede, Kindu Agmas, Demewoz Kefale.

**Formal analysis:** Tigabu Munye Aytenew, Demewoz Kefale.

**Investigation:** Solomon Demis Kebede, Tigabu Munye Aytenew.

**Methodology:** Solomon Demis Kebede, Amare Kassaw, Tigabu Munye Aytenew, Demewoz Kefale.

**Supervision:** Amare Kassaw, Kindu Agmas.

**Validation:** Kindu Agmas, Demewoz Kefale.

**Visualization:** Amare Kassaw.

**Writing – original draft:** Solomon Demis Kebede, Demewoz Kefale.

**Writing – review & editing:** Solomon Demis Kebede.

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
