## [Decision Letter · Decision Letter 0]

2 Sep 2024

PONE-D-23-19848Prevalence of persistent neonatal hypoglycemia and its associated factors in Ethiopia: A systematic review and meta-analysisPLOS ONE

Dear Dr. Demis,

Thank you for submitting your manuscript to PLOS ONE. After careful consideration, we feel that it has merit but does not fully meet PLOS ONE’s publication criteria as it currently stands. Therefore, we invite you to submit a revised version of the manuscript that addresses the points raised during the review process.

We look forward to receiving your revised manuscript.

Kind regards,

Saidul Abrar, MBBS, MPH

Academic Editor

PLOS ONE

Journal requirements: 1. When submitting your revision, we need you to address these additional requirements. Please ensure that your manuscript meets PLOS ONE's style requirements, including those for file naming. The PLOS ONE style templates can be found at https://journals.plos.org/plosone/s/file?id=wjVg/PLOSOne_formatting_sample_main_body.pdf and https://journals.plos.org/plosone/s/file?id=ba62/PLOSOne_formatting_sample_title_authors_affiliations.pdf. 2. Note from Emily Chenette, Editor in Chief of PLOS ONE, and Iain Hrynaszkiewicz, Director of Open Research Solutions at PLOS: Did you know that depositing data in a repository is associated with up to a 25% citation advantage (https://doi.org/10.1371/journal.pone.0230416)? If you’ve not already done so, consider depositing your raw data in a repository to ensure your work is read, appreciated and cited by the largest possible audience. You’ll also earn an Accessible Data icon on your published paper if you deposit your data in any participating repository (https://plos.org/open-science/open-data/#accessible-data) 3. Thank you for stating the following financial disclosure:  [No].  At this time, please address the following queries: a) Please clarify the sources of funding (financial or material support) for your study. List the grants or organizations that supported your study, including funding received from your institution. b) State what role the funders took in the study. If the funders had no role in your study, please state: “The funders had no role in study design, data collection and analysis, decision to publish, or preparation of the manuscript.”c) If any authors received a salary from any of your funders, please state which authors and which funders.d) If you did not receive any funding for this study, please state: “The authors received no specific funding for this work.” Please include your amended statements within your cover letter; we will change the online submission form on your behalf. 4. In the online submission form, you indicated that [The data will be available upon reasonable request.]. All PLOS journals now require all data underlying the findings described in their manuscript to be freely available to other researchers, either 1. In a public repository, 2. Within the manuscript itself, or 3. Uploaded as supplementary information.This policy applies to all data except where public deposition would breach compliance with the protocol approved by your research ethics board. If your data cannot be made publicly available for ethical or legal reasons (e.g., public availability would compromise patient privacy), please explain your reasons on resubmission and your exemption request will be escalated for approval.  5. We note that you have referenced (ie. Demis S et al. [27]) which has currently not yet been accepted for publication. Please remove this from your References and amend this to state in the body of your manuscript: (ie “Demis S et al. [Unpublished]”) as detailed online in our guide for authorshttp://journals.plos.org/plosone/s/submission-guidelines#loc-reference-style  6. Please include captions for your Supporting Information files at the end of your manuscript, and update any in-text citations to match accordingly. Please see our Supporting Information guidelines for more information: http://journals.plos.org/plosone/s/supporting-information.  7. As required by our policy on Data Availability, please ensure your manuscript or supplementary information includes the following:  A numbered table of all studies identified in the literature search, including those that were excluded from the analyses.   For every excluded study, the table should list the reason(s) for exclusion.   If any of the included studies are unpublished, include a link (URL) to the primary source or detailed information about how the content can be accessed.  A table of all data extracted from the primary research sources for the systematic review and/or meta-analysis. The table must include the following information for each study:  Name of data extractors and date of data extraction  Confirmation that the study was eligible to be included in the review.   All data extracted from each study for the reported systematic review and/or meta-analysis that would be needed to replicate your analyses.  If data or supporting information were obtained from another source (e.g. correspondence with the author of the original research article), please provide the source of data and dates on which the data/information were obtained by your research group.  If applicable for your analysis, a table showing the completed risk of bias and quality/certainty assessments for each study or outcome.  Please ensure this is provided for each domain or parameter assessed. For example, if you used the Cochrane risk-of-bias tool for randomized trials, provide answers to each of the signalling questions for each study. If you used GRADE to assess certainty of evidence, provide judgements about each of the quality of evidence factor. This should be provided for each outcome.   An explanation of how missing data were handled.   This information can be included in the main text, supplementary information, or relevant data repository. Please note that providing these underlying data is a requirement for publication in this journal, and if these data are not provided your manuscript might be rejected.  

Reviewers' comments:

Reviewer's Responses to Questions

**Comments to the Author**

1. Is the manuscript technically sound, and do the data support the conclusions?

Reviewer #1: Yes

Reviewer #2: No

2. Has the statistical analysis been performed appropriately and rigorously? 

Reviewer #1: Yes

Reviewer #2: I Don't Know

3. Have the authors made all data underlying the findings in their manuscript fully available?

Reviewer #1: Yes

Reviewer #2: Yes

4. Is the manuscript presented in an intelligible fashion and written in standard English?

Reviewer #1: Yes

Reviewer #2: Yes

5. Review Comments to the Author

Reviewer #1: The manuscript was well written with each section clearly mentioned and detailed except for minor grammatical errors. It will be nice if the authors could also add "limitation of study" section which was obviously missing

Reviewer #2: “Prevalence of persistent neonatal hypoglycemia and its associated factors in Ethiopia: A systematic review and meta-analysis”

Good beginning; yet few areas of improvement and the readers/reviewers concerns / suggestion to be incorporated

1. Good title to capture the reader of different healthcare professionals

2. In the abstract it is mentioned; the sample size was 3686 from the 8 studies but later on it is mentioned 3600? It needs to be clear for the readers

3. Interested background and introduction; keeps the readers indulged in the reading; very good

4. Good rationale for the study and gives the readers novelty

5. PICO format has the biggest concerns: Readers are totally confused at once; sudden change from the hypoglycemia to intra-Ventricular hemorrhage???? Does that align with the title of the study? Turning point from the winners to losers in the eyes of the readers? Unbearable as reader is not happy with the twist; clear the reader what the study is about?

6. Completely opposing from the title as per readers best judgment??? Derailed from the track? I don't know how and why it happened??? Many questions arise. As stated in PICO format, (the following review questions were developed for the search for data. What is the magnitude of IVH in African countries? What are the predictors of IVH in African countries?) Reader is confused, what the outcome is stated and what it should be as per the title of the study???

7. The key terms in the abstract are mentioned as “Prevalence, persistent neonatal hypoglycemia, prolonged neonatal hypoglycemia, Ethiopia (these were the key words as stated earlier in the abstract); suddenly all these are mentioned later on as "The core search terms and phrases were “prevalence”, “incidence”, “epidemiology”, “proportion”, “magnitude”, “burden”, “Intraventricular” and ‘’predictors’’, “risk factors’’, ‘’associated factors’’ and ‘’Africa’’ Or ‘’African countries"???? Huge gap exists

8. time frame in the inclusion criteria need to be mentioned

9. Pakistan; Iran and Israel are from different region (apart from Ethiopia); do these studies align with the inclusion criteria??? Referring to the second Para of the discussion chapter?

10. Referring to Critical appraisal and quality assessment portion ; it would be more appropriate if the examples be given how the criteria was accomplished in the particular study?

11. Strength and limitation of this study depicts 3600 neonates were included??? Does that mean a sample size??? If yes, previously it was mentioned that 3686 neonates were the ample size? Clarity is suggested

6. PLOS authors have the option to publish the peer review history of their article (what does this mean?). If published, this will include your full peer review and any attached files.

Reviewer #1: **Yes: **Fatima Abubakar Ishaq

Reviewer #2: No

---

## [Author Response · Author response to Decision Letter 0]

4 Sep 2024

Authors’ response to reviewers’ comments 

 “Prevalence of persistent neonatal hypoglycemia and its associated factors in Ethiopia: A systematic review and meta-analysis”

Good beginning; yet few areas of improvement and the readers/reviewers concerns / suggestion to be incorporated 

1. Good title to capture the reader of different healthcare professionals

2. In the abstract it is mentioned; the sample size was 3686 from the 8 studies but later on it is mentioned 3600? It needs to be clear for the readers

Authors’ response: revised 

3. Interested background and introduction; keeps the readers indulged in the reading; very good

4. Good rationale for the study and gives the readers novelty 

5. PICO format has the biggest concerns: Readers are totally confused at once; sudden change from the hypoglycemia to intra-Ventricular hemorrhage???? Does that align with the title of the study? Turning point from the winners to losers in the eyes of the readers? Unbearable as reader is not happy with the twist; clear the reader what the study is about?

Authors’ response: revised and we ask for pardon. We made writing errors. 

6. Completely opposing from the title as per readers best judgment??? Derailed from the track? I don't know how and why it happened??? Many questions arise. As stated in PICO format, (the following review questions were developed for the search for data. What is the magnitude of IVH in African countries? What are the predictors of IVH in African countries?) Reader is confused, what the outcome is stated and what it should be as per the title of the study??? 

Authors’ response: yes we did it correct. 

7. The key terms in the abstract are mentioned as “Prevalence, persistent neonatal hypoglycemia, prolonged neonatal hypoglycemia, Ethiopia (these were the key words as stated earlier in the abstract); suddenly all these are mentioned later on as "The core search terms and phrases were “prevalence”, “incidence”, “epidemiology”, “proportion”, “magnitude”, “burden”, “Intraventricular” and ‘’predictors’’, “risk factors’’, ‘’associated factors’’ and ‘’Africa’’ Or ‘’African countries"???? Huge gap exists

Authors’ response: corrected typos errors

8. time frame in the inclusion criteria need to be mentioned

Authors’ response: the articles included were those published from 2020 to 2022. 

9. Pakistan; Iran and Israel are from different region (apart from Ethiopia); do these studies align with the inclusion criteria??? Referring to the second Para of the discussion chapter?

Authors’ response: this is to insight the burden across the globe…not to analyze It is aimed to show the difference in different regions but not part for this review. 

10. Referring to Critical appraisal and quality assessment portion ; it would be more appropriate if the examples be given how the criteria was accomplished in the particular study?

Authors’ response: yes the comment is accepted and given full reference to supplementary file 2. 

11. Strength and limitation of this study depicts 3600 neonates were included??? Does that mean a sample size??? If yes, previously it was mentioned that 3686 neonates were the ample size? Clarity is suggested

Authors’ response: revised and sorry for repeated writing errors. 

12. Data extraction date and authors’ put beneath the table and that was from 20-27 May 2023. 

13. The article by Demis et al., 2022 was accepted for publication and that is why we cited as it was there.

FUNDING INFORMATION:

14. No specific funding was received for this work.

DATA AVAILABILITY

15. Data was uploaded as supporting information.

---

## [Decision Letter · Decision Letter 1]

29 Nov 2024

PONE-D-23-19848R1Prevalence of prolonged transitional neonatal hypoglycemia and associated factors in Ethiopia: A systematic review and meta-analysisPLOS ONE

Dear Dr. Demis,

Thank you for submitting your manuscript to PLOS ONE. After careful consideration, we feel that it has merit but does not fully meet PLOS ONE’s publication criteria as it currently stands. Therefore, we invite you to submit a revised version of the manuscript that addresses the points raised during the review process.

We look forward to receiving your revised manuscript.

Kind regards,

Saidul Abrar

Academic Editor

PLOS ONE

Additional Editor Comments :

Kindly contact staff editor. Thanks

Reviewers' comments:

Reviewer's Responses to Questions

**Comments to the Author**

1. If the authors have adequately addressed your comments raised in a previous round of review and you feel that this manuscript is now acceptable for publication, you may indicate that here to bypass the “Comments to the Author” section, enter your conflict of interest statement in the “Confidential to Editor” section, and submit your "Accept" recommendation.

Reviewer #1: All comments have been addressed

2. Is the manuscript technically sound, and do the data support the conclusions?

Reviewer #1: Yes

3. Has the statistical analysis been performed appropriately and rigorously? 

Reviewer #1: Yes

4. Have the authors made all data underlying the findings in their manuscript fully available?

Reviewer #1: Yes

5. Is the manuscript presented in an intelligible fashion and written in standard English?

Reviewer #1: Yes

6. Review Comments to the Author

Reviewer #1: The observations raised by the reviewers have been well addressed. All aspect of the manuscript (Introduction, aim and objective, methodology, results, discussion and limitation)were all covered approriately

7. PLOS authors have the option to publish the peer review history of their article (what does this mean?). If published, this will include your full peer review and any attached files.

Reviewer #1: **Yes: **Fatima Abubakar Ishaq

---

## [Author Response · Author response to Decision Letter 1]

4 Dec 2024

Dear Reviewers,

We, the authors, have carefully revised the manuscript in response to all of the reviewers' comments, and we have also addressed additional typographical and grammatical revisions to ensure the manuscript is clear and well-structured (R2). In this revised version (R2), we have not identified any new revision requests from the reviewers. However, we took the opportunity to further refine and enhance the manuscript to improve its overall readability, coherence, and clarity.

We believe that publishing this manuscript is important because it addresses critical issues related to neonatal hypoglycemia, a condition with significant clinical implications. The findings from this study could provide valuable insights into improving neonatal care, reducing morbidity, and guiding best practices for early diagnosis and management of hypoglycemia in neonates. Given the increasing burden of neonatal health challenges globally, particularly in resource-limited settings, our research has the potential to contribute to evidence-based strategies that can enhance neonatal outcomes and inform healthcare policies.

We hope this manuscript will contribute meaningfully to the existing body of knowledge and support ongoing efforts to improve neonatal health outcomes worldwide.

Kind regards,

Solomon Demis Kebede (Corresponding Author)

---

## [Editor Report · Decision Letter 2]

9 Dec 2024

PONE-D-23-19848R2Prevalence of prolonged transitional neonatal hypoglycemia and associated factors in Ethiopia: A systematic review and meta-analysisPLOS ONE

Dear Dr. Demis,

Thank you for submitting your manuscript to PLOS ONE. After careful consideration, we feel that it has merit but does not fully meet PLOS ONE’s publication criteria as it currently stands. Therefore, we invite you to submit a revised version of the manuscript that addresses the points raised during the review process.

We look forward to receiving your revised manuscript.

Kind regards,

Saidul Abrar, MBBS, MPH

Academic Editor

PLOS ONE

Journal Requirements:

Additional Editor Comments:

Dear Author,

Kindly Address R1 reviews and submit that as R2 file.Thanks

---

## [Author Response · Author response to Decision Letter 2]

10 Dec 2024

We have revised the manuscript in this submission, focusing primarily on updating the references. As requested, we have uploaded two versions: the manuscript with track changes and the clean version.

---

## [Editor Report · Decision Letter 3]

12 Dec 2024

Prevalence of prolonged transitional neonatal hypoglycemia and associated factors in Ethiopia: A systematic review and meta-analysis

PONE-D-23-19848R3

Dear Author, we are pleased to inform you that your manuscript has been judged scientifically suitable for publication and will be formally accepted for publication once it meets all outstanding technical requirements.

Kind regards,

Saidul Abrar, MBBS, MPH

Academic Editor

PLOS ONE

Additional Editor Comments (optional):

Kindly follow the instructions for further processing.Best of luck.
---

## [Editor Report · Acceptance letter]

17 Dec 2024

PONE-D-23-19848R3 

PLOS ONE

Dear Dr. Demis, 

I'm pleased to inform you that your manuscript has been deemed suitable for publication in PLOS ONE. Congratulations! Your manuscript is now being handed over to our production team.

Kind regards, 

on behalf of

Dr Saidul Abrar 

Academic Editor

PLOS ONE